# Método constructivo para la generación de modelos de comportamiento a partir de reportes de usuario en sistemas *software*

**Alejandro Aunión**
Departamento de Informática y Estadística
Universidad Rey Juan Carlos
Móstoles, 28933, Madrid, Spain
a.aunion.2021@alumnos.urjc.es

**Javier Yuste**\*
Departamento de Informática y Estadística
Universidad Rey Juan Carlos
Móstoles, 28933, Madrid, Spain
javier.yuste@urjc.es

**Eduardo G. Pardo**
Departamento de Informática y Estadística
Universidad Rey Juan Carlos
Móstoles, 28933, Madrid, Spain
eduardo.pardo@urjc.es

## Abstract

La calidad de los sistemas *software* tiende a deteriorarse con el tiempo a medida que estos evolucionan. Crear casos de prueba apropiados es una tarea compleja, que puede llevar a correcciones incompletas o incluso a la introducción de nuevos fallos si no se realiza correctamente. El área de *Model-Based Testing* se centra en el diseño de casos de prueba de calidad de manera automática o semi-automática. Para ello, es necesario disponer de modelos que representen el comportamiento del sistema que se quiere probar. En este trabajo, se estudia el problema de la inferencia de modelos de comportamiento, un problema de optimización multi-objetivo que busca construir modelos de comportamiento de calidad. En primer lugar, se identifica un conjunto de soluciones triviales en el espacio de búsqueda del problema estudiado. Para recorrer el espacio de búsqueda entre dichas soluciones triviales, se propone un método constructivo semi-voraz multi-arranque, cuyo rendimiento se compara con el de los métodos propuestos en la literatura. Los resultados obtenidos muestran que la estrategia propuesta tiene potencial para generar un conjunto de soluciones no dominadas de calidad. Finalmente, se describen varias líneas de trabajo futuro para mejorar la eficiencia del método propuesto.

## 1. Introducción

A medida que los sistemas *software* crecen y evolucionan, su calidad tiende a deteriorarse [11]. El mantenimiento del *software* se centra en mantener la capacidad del sistema para proveer el servicio para el que está diseñado [10], incluyendo actividades destinadas a corregir fallos, mejorar el rendimiento o adaptar el producto a cambios en el entorno [2]. Estas tareas suelen suponer hasta un 80 % de los costes totales del sistema [4], principalmente debido al esfuerzo necesario para comprenderlo [12]. Crear casos de prueba apropiados es una tarea compleja, y un diseño imperfecto de estos puede llevar a correcciones incompletas o incluso a la introducción de nuevos fallos en el código [13].

---

\*Corresponding author

XVI XVI Congreso Español de Metaheurísticas, Algoritmos Evolutivos y Bioinspirados (maeb 2025).

En ingeniería del *software*, *Model-Based Testing* (MBT) se refiere al conjunto de actividades que buscan diseñar casos de prueba para un sistema a partir de modelos del mismo, ya sea de manera manual, automática o semi-automática [3]. Como se puede intuir, la calidad de los modelos del sistema es un aspecto de vital importancia para las actividades de MBT [15]. Para estos fines, existen diferentes tipos de modelos que pueden ser de utilidad, como los modelos de comportamiento. Estos modelos se pueden representar mediante máquinas de estado que reflejan una secuencia de actividades o comportamientos del sistema [7].

Para inferir modelos de comportamiento de manera automática, diversos autores han propuesto la utilización de metaheurísticas [15]. En un trabajo reciente [9], los autores propusieron una aproximación para generar modelos de comportamiento de manera automática a partir de errores reportados por usuarios. En particular, los autores proponen abordar la inferencia de modelos de comportamiento como un problema de optimización multi-objetivo en el que, dado un conjunto de trazas (reportes de usuarios procesados) como entrada, se busca generar modelos que optimizan tres objetivos distintos. Para abordar este problema de optimización, los autores propusieron la utilización de tres métodos ampliamente conocidos en la literatura: NSGA-II [6], NSGA-III [5] y MOEA/D [14].

Este trabajo se centra en el problema de la inferencia de modelos de comportamiento propuesto por Guizzo y compañía [9]. En primer lugar, se identifica un conjunto de soluciones triviales en el espacio de búsqueda del problema estudiado. A continuación, se propone un método constructivo semi-voraz multi-arranque, cuyo rendimiento se evalúa y compara con el de los métodos propuestos en la literatura.

El resto del artículo se estructura de la siguiente manera. En la Sección 2, se define el problema estudiado. En la Sección 3, se describe el método propuesto en este trabajo. En la Sección 4, se reportan los resultados obtenidos en la experimentación realizada. Finalmente, en la Sección 5, se exponen las conclusiones obtenidas y se discuten algunas líneas de trabajo futuro.

## 2. Definición del problema

El problema estudiado en este trabajo es el problema de la inferencia de modelos de comportamiento a partir de reportes de usuarios, propuesto por Guizzo y compañía [9]. Este es un problema de optimización multi-objetivo que busca encontrar un modelo de comportamiento ($m$) capaz de reproducir los errores reportados en un conjunto de trazas de entrada ($W$). Un conjunto de trazas de entrada $W$ es una instancia del problema. Cada traza de entrada es un autómata que representa la secuencia de actividades que reproducen un error reportado por un usuario. Por lo tanto, una instancia para el problema es un conjunto de autómatas. En particular, una traza es un autómata definido formalmente como una tupla $(Q, \Sigma, \delta, q_0, F)$, donde:

- $Q$ es un conjunto finito de estados.
- $\Sigma$ es el alfabeto de entrada. Es decir, el conjunto de actividades que se pueden realizar y que se representan mediante transiciones entre estados.
- $\delta : Q \times \Sigma \to Q$ es la función de transición, que une dos estados $q_a, q_b \in Q$ mediante una palabra $\sigma \in \Sigma$.
- $q_0 \in Q$ es el estado inicial.
- $F \subseteq Q$ es el conjunto de estados finales o de aceptación.

En la Figura 1, se muestran dos trazas de ejemplo. Como se puede observar, se trata de autómatas en los que existen un conjunto de estados y un conjunto de transiciones entre estados. La secuencia de transiciones mostrada representa una secuencia de actividades que provocan un fallo en el sistema. En este caso, la primera traza, representada en la Figura 1a, contiene las siguientes actividades: "Abrir Firefox", "Cargar página" y "Cerrar Firefox". La segunda traza, representada en la Figura 1b, contiene las siguientes actividades: "Abrir Firefox", "Búsqueda" y "Recargar ventana". En ambas soluciones existe un estado inicial ($q_0$) y un estado final ($q_3$). Nótese que todas las trazas de entrada son autómatas sin ciclos y donde para todos los estados el grado máximo de entrada y de salida es uno.

Una solución para el problema se representa mediante un modelo $m$, un autómata definido formalmente como una tupla $(Q, \Sigma, \delta, q_0, F)$, de igual modo que las trazas. De hecho, cada una de las

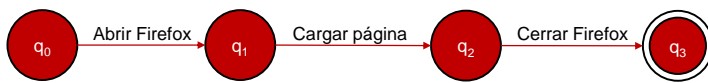

(a) Traza de entrada para una instancia del problema con tres transiciones: "Abrir Firefox", "Cargar página" y "Cerrar Firefox". El estado inicial es $q_0$ y hay un estado final: $q_3$.

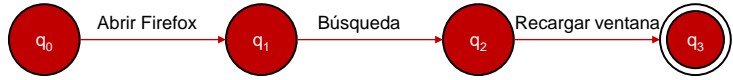

(b) Traza de entrada para una instancia del problema con tres transiciones: "Abrir Firefox", "Búsqueda" y "Recargar ventana". El estado inicial es $q_0$ y hay un estado final: $q_3$.

Figura 1: Dos trazas de entrada de ejemplo para una instancia del problema. Las trazas se representan mediante autómatas que modelan la secuencia de actividades reportada por un usuario y que provoca un fallo en el sistema.

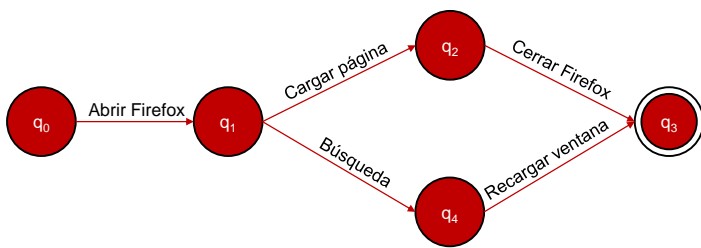

Figura 2: Ejemplo de una solución para el problema, obtenida mediante la unión de las dos trazas de ejemplo mostradas anteriormente en la Figura 1. El estado inicial es $q_0$ y hay un estado final: $q_3$.

trazas de entrada es una posible solución. En la Figura 2, se muestra una solución de ejemplo para el problema obtenida mediante la unión de las dos trazas de la Figura 1. Nótese que las soluciones, al contrario que las trazas de entrada, pueden contener ciclos y no tienen limitaciones con respecto al grado máximo de entrada y salida de sus estados.

Para evaluar la calidad de las soluciones generadas, en la literatura se proponen tres funciones objetivo [9]. La primera de las funciones objetivo se conoce como subestimación (UA, del inglés *Under-Approximation*). Esta métrica mide el número de trazas de entrada que la solución no representa. Una solución representa una traza de entrada si la secuencia de actividades de dicha traza es una palabra reconocida por el autómata de la solución. Formalmente, la subestimación se define como se muestra en la Ecuación 1, donde $m$ es el modelo a evaluar y $W$ es el conjunto de trazas de entrada. El objetivo es minimizar el valor obtenido. Por lo tanto, una solución óptima sería aquella que representara a todas las trazas de entrada.

$$\downarrow \mathrm{UA}(m, W) = \sum_{w \in W} \left\{ \begin{array}{ll} 0 & \text{si } w \in m \\ 1 & \text{en caso contrario,} \end{array} \right. \tag{1}$$

La segunda de las funciones objetivo se conoce como sobrestimación (OA, del inglés *Over-Approximation*). Esta métrica mide el número de caminos válidos representados por el modelo $m$ que no están contenidos en ninguna de las trazas de entrada. Aquí, un cámino válido es una secuencia de transiciones aceptada por el autómata de la solución. Formalmente, la sobrestimación se define como se muestra en la Ecuación 2, donde $m$ es el modelo a evaluar, $W$ es el conjunto de trazas de entrada y $P$ es el conjunto de caminos válidos en $m$. El objetivo es minimizar el valor obtenido. Por lo tanto, una solución óptima sería aquella que no tuviera ningún camino válido que no estuviese contenido en alguna traza de entrada. Es decir, aquel modelo que no representa secuencias de actividades que no han sido reportadas. Nótese que, por el esfuerzo computacional que supone

evaluar este objetivo para soluciones de gran tamaño, los autores limitan la evaluación a caminos de longitud cuatro. Es decir, $P$ en realidad está formado solo por los caminos válidos de la solución $m$ que tienen cuatro transiciones o menos.

$$\downarrow \mathrm{OA}(P, W) = \sum_{p \in P} \left\{ \begin{array}{ll} 0 & \text{si } p \in W \\ 1 & \text{en caso contrario} \end{array} \right. \tag{2}$$

La tercera de las funciones objetivo es el tamaño del modelo, es decir, el número de estados. Formalmente, el tamaño se define como se muestra en la Ecuación 3, donde $m$ es el modelo a evaluar y $Q$ es el conjunto de estados del modelo. De nuevo, el objetivo es minimizar el valor obtenido. Por lo tanto, una solución es mejor cuantos menos estados tiene, ya que el modelo es más fácil de entender por parte de los desarrolladores y, por lo tanto, es más fácil generar casos de prueba correctos y entendibles a partir del mismo.

$$\downarrow \mathrm{TAMA\tilde{N}O}(m) = |Q| \tag{3}$$

Como se puede intuir, el objetivo es generar modelos encontrando un equilibrio entre el número de trazas representadas y el tamaño del modelo, penalizando las secuencias de actividades que no reflejan comportamientos reportados en las trazas. Finalmente, en la literatura se proponen dos restricciones por cuestiones prácticas: los valores de tamaño y sobrestimación tienen que ser menores de 500. Esto es así para evitar construir modelos que no son de utilidad en la práctica para los desarrolladores.

## 3. Propuesta algorítmica

Como se describió en la definición del problema (Sección 2), las trazas de entrada se modelan mediante autómatas, del mismo modo que las soluciones. Por ende, se puede considerar cada traza como una posible solución al problema. Estas soluciones triviales representan al menos una traza del conjunto de entrada. Por otro lado, la unión de todas las trazas en un mismo autómata resultaría en otra solución trivial, una solución que representaría a todas las trazas de entrada. Como se puede intuir, las soluciones triviales obtenidas de las trazas iniciales tendrían mala subestimación y poco tamaño, mientras que la solución trivial resultante de la unión de todas las trazas de entrada tendría un valor óptimo de subestimación y un valor pésimo de tamaño. Entre estos dos extremos existe un espacio de búsqueda que es posible explorar mediante la adición de transiciones a las soluciones triviales del primer conjunto. En la Figura 3, se ilustra este espacio de búsqueda. A la izquierda, se representan las soluciones triviales que representan las trazas de entrada. A la derecha, se representa la solución trivial resultante de la unión de todas las trazas de entrada. Entre ambos extremos, se representan distintos niveles en el espacio de búsqueda. Cada nivel $N_i$ representa el conjunto de soluciones que se pueden obtener mediante la adición de $i$ transiciones a las soluciones triviales del nivel $N_0$.

En este trabajo se propone un método constructivo semi-voraz multi-arranque que genera soluciones recorriendo el espacio de búsqueda descrito anteriormente. El método está basado en la metodología GRASP (del inglés *Greedy Randomized Adaptive Search Procedure*) propuesta por Thomas Feo y Mauricio Resende en 1989 [8]. En el Algoritmo 1 se muestra el pseudocódigo del método propuesto. Este método recibe como entrada cuatro parámetros: el conjunto de trazas ($W$), el conjunto de transiciones posibles ($\Sigma$), un tiempo límite ($t_{max}$), un criterio voraz ($g()$) y un valor $\alpha$ entre 0 y 1. En primer lugar, el método inicializa una lista de soluciones no dominadas, el archivo (línea 3). A continuación, se escoge una traza al azar y construye una solución que represente dicha traza (línea 6). A esta solución inicial ($sol$) se le irán añadiendo transiciones posteriormente, de manera iterativa, mientras que la solución actual no supere un tamaño de 500 estados y no se supere el tiempo límite (líneas 9-18). En cada iteración, se construye una lista de candidatos ($CL$) con todos los candidatos posibles que se pueden añadir a la solución (línea 10). En este caso, la lista de candidatos incluye la adición de cada transición del alfabeto ($\Sigma$) a partir de cada uno de los estados en la solución actual ($Q$). Por ende, el tamaño de la $CL$ en cada iteración es $|Q| \cdot |\Sigma|$. A continuación, se calculan la calidad del mejor ($g_{min}$) y del peor ($g_{max}$) candidato de la $CL$ y un umbral de aceptación ($th$) que depende de $g_{min}$, $g_{max}$ y el valor de $\alpha$ recibido por parámetro (líneas 11-13). Con estos valores, se construye una lista de candidatos restringida ($RCL$) que contiene los candidatos con una calidad mejor que la del umbral establecido (línea 14). Finalmente, se escoge un candidato al azar de la $RCL$ y se

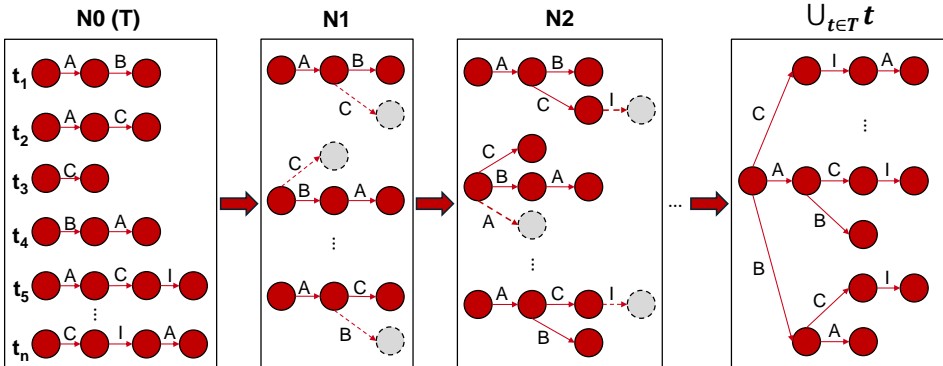

Figura 3: Ilustración del espacio de búsqueda existente entre los dos tipos de soluciones triviales identificados para el problema estudiado. A la izquierda, se representan las soluciones triviales que representan las trazas de entrada. A la derecha, se representa la solución trivial resultante de la unión de todas las trazas de entrada. Cada nivel $N_i$ representa el conjunto de soluciones que se pueden obtener mediante la adición de $i$ transiciones a las soluciones triviales del nivel $N_0$.

añade a la solución (líneas 15-16). La solución obtenida se intenta añadir a la lista de soluciones no dominadas (líneas 17). Dicha solución solo se guarda en la lista si no está dominada, y todas aquellas soluciones dominadas por esta se eliminan del archivo. El archivo no tiene un tamaño máximo. Este proceso se repite mientras que no se supere el tiempo límite (líneas 5-19). Al terminar, el método devuelve un archivo con las soluciones no dominadas que se han construido (línea 20).

---

**Algorithm 1** Constructivo voraz multi-arranque

---

1: **procedure** CONSTRUCTIVO($W$, $\Sigma$, $t_{max}$, $g()$, $\alpha$)
2:     $start \leftarrow$ CPUT()
3:     $Archive \leftarrow \emptyset$
4:     $t \leftarrow$ CPUT()
5:     **while** $start - t \leq t_{max}$ **do**
6:         $sol \leftarrow$ SELECCIÓNALEATORIA($W$)
7:         $Archive \leftarrow$ AÑADE($Archive$, $sol$)
8:         $t \leftarrow$ CPUT()
9:         **while** TAMAÑO($sol$) $< 500$ & OA($sol$) $< 500$ & $start - t \leq t_{max}$ **do**
10:             $CL \leftarrow Q \times \Sigma$
11:             $g_{max} \leftarrow$ máx$\{g(c) \mid c \in CL\}$
12:             $g_{min} \leftarrow$ mín$\{g(c) \mid c \in CL\}$
13:             $th \leftarrow g_{max} - \alpha \cdot (g_{max} - g_{min})$
14:             $RCL \leftarrow \{c \in CL \mid g(c) \leq th\}$
15:             $c \leftarrow$ SELECCIÓNALEATORIA($RCL$)
16:             $sol \leftarrow sol \cup \{c\}$
17:             $Archive \leftarrow$ AÑADE($Archive$, $sol$)
18:         **end while**
19:     **end while**
20:     **return** $Archive$
21: **end procedure**

---

Como se puede observar, el parámetro $\alpha$ controla la voracidad o aleatoriedad del método. Cuanto más cercano a 0 es su valor, más aleatoria es la selección de candidatos. Un valor $\alpha = 0$ implica que los candidatos se escogen al azar, mientras que un valor $\alpha = 1$ implica que los candidatos se escogen de manera voraz. Este parámetro se ajusta de manera experimental en la Sección 4.2.

Para el criterio voraz del método propuesto ($g()$), se podría utilizar alguna de las tres funciones objetivo descritas en la Sección 2. Sin embargo, no todas ellas son adecuadas. En primer lugar, el tamaño no es adecuado porque cualquier candidato añade exactamente un estado. Por lo tanto, ofrecería un espacio de búsqueda plano. La sobrestimación podría ser un buen criterio, ya que evitaría

añadir transiciones que no van a llevar a representar nuevas trazas. Sin embargo, por el esfuerzo computacional que conlleva evaluar este objetivo, no es práctico. Por lo tanto, se ha utilizado el valor de la subestimación como criterio voraz. De este modo, se busca elegir candidatos que llevan a la solución a representar nuevas trazas.

En el esquema original propuesto para la metodología GRASP, además de un método constructivo, los autores proponían utilizar una búsqueda local [8]. En este contexto, no sería adecuado implementar una búsqueda local basada en un operador de adición de una transición, ya que es probable encontrar soluciones para las que ninguno de los candidatos (vecinos, en el caso de la búsqueda local) suponen una mejora. Para implementar una búsqueda local, sería necesario emplear un operador de vecindad más complejo, como adición de múltiples transiciones o eliminación de estas. El diseño de operadores de vecindad más complejos y su implementación se explorará en el futuro.

## 4. Resultados Experimentales

En esta sección, se presentan algunos de los experimentos realizados durante el desarrollo de este trabajo. Todos los experimentos se han realizado en un servidor con procesador AMD EPYC 7282 con 8 cores, 32GB de memoria RAM y un sistema operativo Ubuntu 20.04.1 LTS. Todos los métodos se han implementado en el lenguaje de programación Python 3.12. En todos los experimentos se ha utilizado una instancia seleccionada aleatoriamente del conjunto propuesto por la literatura [9], *Calendar*, que tiene un conjunto de 3220 trazas de entrada, con una media, mediana y desviación estándar de 5,42, 5 y 3,27 estados por traza, respectivamente.

El resto de la sección se estructura como sigue. En la Sección 4.1, se describen los indicadores de calidad utilizados. En la Sección 4.2, se configura experimentalmente el valor del parámetro $alpha$ del método propuesto. Finalmente, en la Sección 4.3 se presenta una comparativa del rendimiento del método propuesto con los métodos propuestos en la literatura.

### 4.1. Indicadores de calidad

Para evaluar la calidad de las soluciones en problemas de optimización multi-objetivo, en la literatura se suele recomendar utilizar Indicadores de Calidad (QIs, del inglés *Quality Indicators*) apropiados en lugar de comparar objetivos individuales de forma aislada [1]. Sin embargo, seleccionar los QIs apropiados no es trivial, y no hay acuerdo en problemas de optimización de ingeniería del software con respecto a cuáles hay que utilizar [1]. En general, los QIs pueden considerar cuatro aspectos diferentes de los conjuntos de soluciones a evaluar: convergencia, dispersión, uniformidad y cardinalidad. Sin embargo, ningún QI cubre completamente todos los aspectos. Por ello, y siguiendo las sugerencias de la literatura [1], se ha decidido utilizar tres QIs que, en conjunto, evalúan completamente todos los aspectos de calidad deseados: Hipervolumen (HV), *Inverted Generational Distance* (IGD+) y número de soluciones no dominadas (PFS, del inglés *Pareto Front Size*).

El indicador HV evalúa la convergencia, dispersión y uniformidad de las soluciones. Cuanto mayor es su valor, mejor es el conjunto de soluciones evaluado. IGD+ mide la proximidad de las soluciones del conjunto evaluado a un conjunto de referencia $R$. Cuanto menor es su valor, mejor es la calidad del conjunto de soluciones evaluado. Finalmente, PFS mide el número de soluciones no dominadas en el conjunto. Cuanto mayor es su valor, mejor es la calidad del conjunto.

Como se puede notar, algunos de los QIs mencionados necesitan disponer de un frente de referencia $R$. Idealmente, el conjunto de referencia sería el frente de Pareto óptimo. Sin embargo, no siempre se conoce. En este caso, en todos los experimentos se utiliza un frente aproximado obtenido mediante la combinación de las soluciones reportadas por cada uno de los métodos que se están comparando en el experimento en cuestión, como es común en la literatura [1].

### 4.2. Parametrización del método propuesto

En el método propuesto, descrito en el Algoritmo 1, se reciben varios parámetros de entrada. Entre ellos, se encuentra $\alpha$, un parámetro que controla el grado de aleatoriedad del método. En esta sección, se realiza un análisis del rendimiento del método propuesto con diferentes valores de $\alpha$. En particular, se ejecuta una vez el algoritmo con cada uno de los siguientes valores: 1; 0,75; 0,5; 0,25; y 0. Es

| Valor de $\alpha$ | PFS | HV | IGD+ |
|---|---|---|---|
| 1,00 | **43,00** | **0,59** | **0,00** |
| 0,75 | **43,00** | **0,59** | **0,00** |
| 0,50 | 43,00 | 0,58 | <0,01 |
| 0,25 | 43,00 | 0,58 | 0,09 |
| 0,00 | 1,00 | 0,00 | 0,73 |

Tabla 1: Evaluación de los indicadores de calidad HV, PFS e IGD+ para el método propuesto con diferentes valores del parámetro de entrada $\alpha$. Para cada indicador de calidad, se destaca el mejor resultado obtenido en la comparativa con fuente negrita.

importante destacar que todas las versiones empiezan desde las mismas soluciones de partida (línea 6 del Algoritmo 1). Además, se establece un tiempo límite de una hora para todos los métodos.

En la Tabla 1, se muestran los resultados obtenidos con los distintos valores de $\alpha$. Como se puede observar, existe una diferencia notable entre una configuración aleatoria ($\alpha = 0$) y el resto de configuraciones. Probablemente, esto se debe a que es altamente improbable añadir transiciones que representan trazas de entrada de manera aleatoria. En contraste, cuando se utiliza el criterio voraz propuesto, aunque sea con un valor pequeño de $\alpha$, el método es capaz de añadir transiciones que aumentan el número de trazas representadas. La poca diferencia existente entre las diferentes configuraciones que utilizan un valor de $\alpha$ mayor que cero podría indicar que no es frecuente encontrar candidatos que mejoren la subestimación en más de una unidad. Si todos los candidatos mejoran el valor del criterio voraz en cero o una unidad, entonces cualquiera de los candidatos que mejoran la solución en una unidad será seleccionado siempre y cuando el valor de $\alpha$ sea mayor a cero. La pequeña mejora en calidad que se puede observar en las configuraciones cuyo valor de $\alpha$ es mayor que 0,50 podría indicar que se ha encontrado algún candidato que mejoraba la calidad de la solución en más de una unidad, y que una configuración más voraz ha llevado al método a priorizar esos candidatos. Por estos motivos, se decide configurar el método propuesto con un valor de $\alpha = 1$, dotándole de un comportamiento completamente voraz.

### 4.3. Comparativa con el estado del arte

En un trabajo reciente, Guizzo y compañía [9] propusieron tres métodos evolutivos ampliamente conocidos para resolver el problema descrito en la Sección 2: NSGA-II [6], NSGA-III [5] y MOEA/D [14]. Además, los autores propusieron varios operadores de cruce y mutación. Por cuestiones de espacio, no detallamos aquí los operadores propuestos en el trabajo citado ni describimos los métodos evolutivos mencionados. En su lugar, redirigimos al lector a los trabajos originales.

En esta sección, se realiza una comparativa del rendimiento del método propuesto en este trabajo (con $\alpha = 1$, como se describe en la Sección 4.2) y de los métodos del estado del arte, propuestos por Guizzo y compañía [9]. Todos los métodos se han ejecutado una vez en la misma plataforma, tal y como se describe al comienzo de la Sección 4, con un tiempo límite de sesenta minutos. Para los métodos propuestos por Guizzo y compañía, se han empleado las implementaciones realizadas por los mismos autores, disponibles públicamente [2].

En la Tabla 2, se muestran los resultados obtenidos en la comparación del método propuesto en este trabajo (C) con los métodos propuestos en el estado del arte. Como se puede observar, C muestra un rendimiento inferior a los métodos propuestos en la literatura. En particular, el conjunto de 43 soluciones no dominadas encontrado tiene peor valor de HV y está más alejado del frente aproximado que los métodos de la literatura. De entre estos últimos, se puede observar que NSGA-II es el método que obtiene un conjunto de soluciones de mayor calidad. Este comportamiento, además, coincide con lo expuesto por los autores originales, que obtuvieron mejores resultados con NSGA-II que con los otros métodos.

En las Figura 4, se muestran en el espacio de objetivos los conjuntos de soluciones no dominadas obtenidas por cada uno de los métodos. En particular, se muestran los conjuntos de soluciones no

---

[2]https://github.com/SOLAR-group/ModelInference

| Método | PFS | HV | IGD+ |
|---|---|---|---|
| C ($\alpha = 1$) | 43,00 | 0,15 | 0,45 |
| NSGA-II [6] | **269,00** | **0,62** | **<0,01** |
| MOEA/D [14] | 66,00 | 0,44 | 0,13 |
| NSGA-III [5] | 39,00 | 0,34 | 0,18 |

Tabla 2: Evaluación de los indicadores de calidad HV, PFS e IGD+ para los conjuntos de soluciones obtenidos por el método propuesto en este trabajo (C) y los métodos propuestos en la literatura [9] (NSGA-II [6], NSGA-III [5] y MOEA/D [14]). Para cada indicador de calidad, se destaca el mejor resultado obtenido en la comparativa con fuente negrita.

dominadas considerando únicamente los objetivos de subestimación y tamaño (Figura 4a), considerando únicamente los objetivos de subestimación y sobrestimación (Figura 4b) y considerando únicamente los objetivos de sobrestimación y tamaño (Figura 4c). Como se puede observar, el método NSGA-II es capaz de obtener soluciones con mejor subestimación que los demás y, además, con poca sobrestimación. Con respecto al método propuesto en este trabajo, C, se puede observar que es capaz de generar soluciones no dominadas, próximas entre sí en el espacio de objetivos, en las que el tamaño aumenta progresivamente a medida que se mejora la subestimación. Esto refleja el procedimiento diseñado, en el que se parte de una solución trivial con poco tamaño y mala subestimación (que estaría en la esquina inferior derecha de la Figura 4a) y se avanza hacia una solución trivial con pésimo tamaño y óptima subestimación (que estaría representada en la esquina superior izquierda de la Figura 4a). Probablemente, con un tiempo de ejecución más largo, el método propuesto sería capaz de encontrar soluciones no dominadas y bien distirbuidas a lo largo del espacio de objetivos, de extremo a extremo.

## 5. Conclusiones y trabajos futuros

En este trabajo, se ha estudiado el problema de optimización multi-objetivo de la inferencia de modelos de comportamiento a partir de reportes de usuario. En este contexto, se ha identificado la existencia de dos tipos de soluciones triviales para el problema: las soluciones que representan alguna de las trazas de entrada y la solución que resulta de la unión de todas las trazas de entrada. El método propuesto, un método constructivo semi-voraz multi-arranque, explora el espacio de búsqueda entre los conjuntos de soluciones triviales identificados. En particular, el método propuesto parte de una solución trivial del primer conjunto y añade transiciones de manera iterativa.

Como se ha mostrado, el rendimiento del método propuesto es actualmente inferior al de los métodos propuestos en el estado del arte considerando varios indicadores de calidad. Sin embargo, el método muestra potencial para obtener un frente de soluciones no dominadas a lo largo del espacio de objetivos que se muestre competitivo con el estado del arte.

En vista de los resultados obtenidos, se han identificado varias áreas de mejora que podrían mejorar la eficiencia del método propuesto. En primer lugar, la implementación de una evaluación incremental de las soluciones permitiría al método recorrer el espacio de búsqueda en menos tiempo. En segundo lugar, una identificación de candidatos prometedores reduciría el espacio de búsqueda, acelerando el método. Además, sería beneficioso evaluar candidatos en vista a posibles mejoras en próximas iteraciones, en vez de tener en cuenta solo el siguiente paso. En tercer lugar, sería adecuado diseñar e implementar una búsqueda local basada en un operador distinto al utilizado en el constructivo. Finalmente, sería deseable que el método propuesto fuera capaz de partir de diferentes puntos repartidos en el espacio de objetivos, en vez de partir de un extremo para avanzar hacia el otro. De esta manera, el rendimiento del método sería mejor en condiciones en las que el tiempo límite no permite completar una iteración.

## Agradecimientos

Este trabajo ha sido parcialmente financiado por los proyectos PID2021-125709OA-C22 y PID2021-126605NB-I00, de MCIN/AEI/10.13039/501100011033 y "ERDF A way of making Europe"; por el

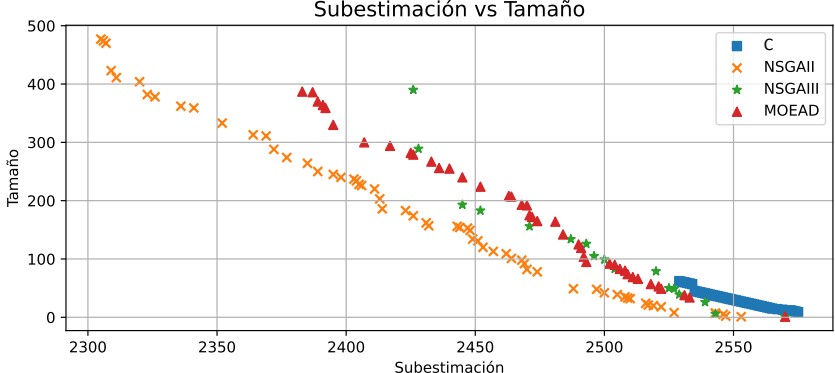

(a) Representación de las soluciones no dominadas obtenidas por cada método considerando los objetivos de subestimación y tamaño.

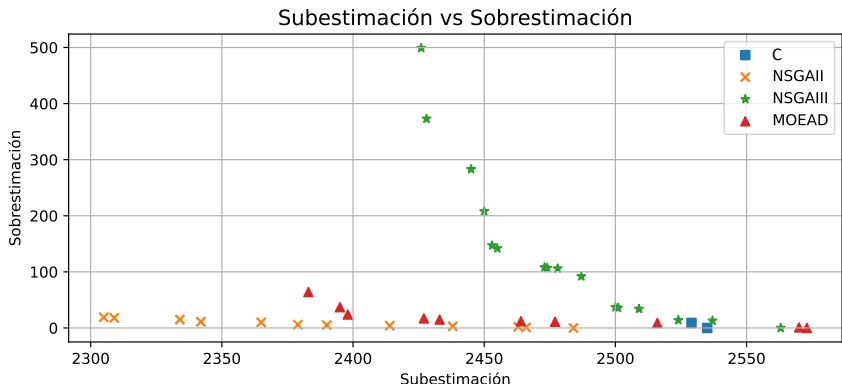

(b) Representación de las soluciones no dominadas obtenidas por cada método considerando los objetivos de subestimación y sobrestimación.

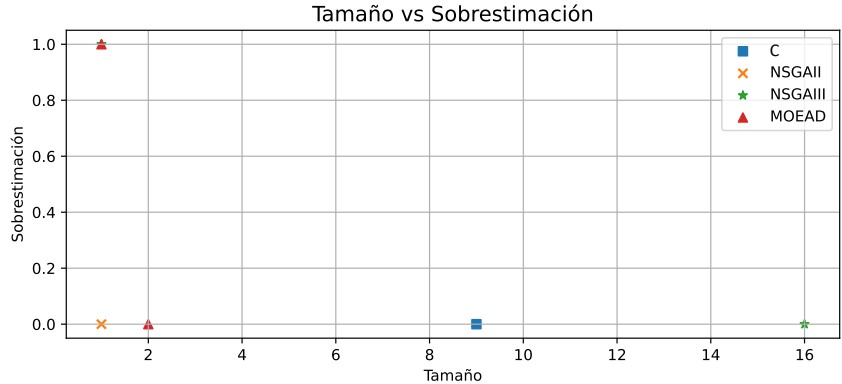

(c) Representación de las soluciones no dominadas obtenidas por cada método considerando los objetivos de tamaño y sobrestimación.

Figura 4: Representación de las soluciones no dominadas obtenidas por cada método (C, NSGA-II [6], NSGA-III [5] y MOEA/D [14]) considerando los distintos objetivos. Por cada par de objetivos, para facilitar la visualización, solo se representan las soluciones no dominadas considerando únicamente ese par de objetivos.

proyecto CIAICO/2021/224 de la Generalitat Valenciana; por el proyecto M3693 de la convocatoria "Adquisición de equipamiento para Altas Capacidades Computación"; por la beca de colaboración 24CO1/001831; por la "Cátedra de Innovación y Digitalización Empresarial entre Universidad Rey Juan Carlos y Second Episode" (Ref. ID MCA06); por la Comunidad Autónoma de Madrid, CIRMA-CM (Ref. TEC-2024/COM-404); y por el Ministerio para la Transformación Digital y de la Función Pública, Concesión TSI-100930-2023-3 (MCA07).

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
