# OpenReview forum: "Método constructivo para la generación de modelos de comportamiento a partir de reportes de usuario en sistemas software"
_MAEB/2025/Congreso — MAEB 2025_

### Official Review · Reviewer_sbCS · 2025-03-17
**Un trabajo con aspectos de interés que requiere de mejoras metodológicas y de presentación**

**Rating:** 3
**Confidence:** 4

**Review:**

El problema considerado parece atractivo y relevante. Se echa en falta, no obstante, un mayor esfuerzo de formalización del mismo. Se define la entrada como un conjunto de autómatas finitos, pero no está claro si realmente la entrada sería una secuencia de estados (lo que sería técnicamente una traza) o cualquier secuencia de estados producida por cualquiera de los autómatas de entrada. Del mismo modo, en la línea 79 se indica que la solución buscada debe representar las trazas de entrada, pero no se formaliza esta noción. Si la entrada consiste en secuencias (cadenas de un lenguaje) puede claramente buscarse un autómata que las reconozca; por otra parte, si la entrada es un conjunto de autómatas, cada uno produciendo un lenguaje L_i, ¿se busca un autómata que reconozca la unión de los L_i y solo esas cadenas? Igualmente en la línea 87 se habla de "caminos válidos". ¿Qué es un camino válido? ¿Una cadena perteneciente al lenguaje generado por el autómata de salida? Sería necesario también clarificar si hay alguna restricción sobre los autómatas buscados (presencia/ausencia de ciclos, grado máximo de entrada o salida, débilmente/fuertemente conexo o no conexo, etc.)

Metodológicamente, no está clara la relevancia de los resultados. No se indica el número de ejecuciones del algoritmo, no se proporcionan medidas de dispersión alrededor de la media, ni se realizan tests estadísticos para ver si las diferencias son significativas. Es difícil entonces determinar si las conclusiones obtenidas son experimentalmente sólidas.

Otros comentarios:
- La observación sobre la búsqueda local en la línea 159 sugiere que la vecindad que debe considerarse no puede estar definida sobre la base de añadir/eliminar una transición, y debe emplearse un operador de vecindad más complejo (añadir y eliminar varias transiciones, quizás con una cierta probabilidad).
- Dado que no se ha empleado el tiempo de cómputo como un factor de estudio, no es necesario indicar el hardware empleado.

---

### Official Review · Reviewer_Tzii · 2025-03-17
**La propuesta quizás es prematura, con más puntos débiles que fuertes.**

**Rating:** 2
**Confidence:** 4

**Review:**

El artículo propone un método tipo GRASP para la generación de modelos de comportamiento en el testeo basado en modelos para software. Se trata de una aproximación multiobjetivo (3 objetivos) basado en GRASP.

El punto fuerte es que el método propuesto es más simple que los algoritmos evolutivos contra los que se compara.

Los puntos débiles son más numerosos:

- Se usa una versión muy simple de GRASP que implementa únicamente la fase constructiva. La justificación de no añadir la fase de mejora local debe explicarse y justificarse de forma más contundente.

- Los resultados obtenidos son bastante pobres, ya que el método propuesto es claramente rebasado por los usados en la comparación.

- El límite de 60m impuesto ¿qué efecto tiene sobre el método propuesto? ¿y sobre el resto? ¿Aumentar el tiempo tiene un efecto positivo sobre los métodos? ¿Para alguno en particular?

- No se ha incluido ningún método no evolutivo o basado en búsqueda en la comparación. ¿los hay? En ese caso, ¿el método propuesto los supera?

---

### Official Review · Reviewer_7y9t · 2025-03-19
**Trabajo preliminar con resultados muy mejorables sobre un problema interesante**

**Rating:** 3
**Confidence:** 5

**Review:**

El trabajo estudia la generación de modelos de comportamiento de programas informáticos mediante la optimización Multi-objetivo. Contar con el modelo de comportamiento de un SW es fundamental para la generación automática de casos de prueba de calidad, algo necesario para la detección de fallos efectiva en el SW. Se propone un método GRASP para la solución del problema, cuyos resultados se comparan con el estado del arte.

Los modelos se generan utilizando autómatas finitos que aceptan trazas de entradas en los que se ha reportado un error por parte de un usuario. El problema estudiado ha sido ya resuelto en la literatura con tres algoritmos evolutivos multi-objetivo, y el método propuesto no es capaz de mejorarlos, ya que aunque en términos de calidad de las soluciones se encuentra cerca de estos algoritmos, sólo es capaz de encontrar soluciones en una zona del frente de Pareto, lo que propicia que el valor de las métricas de calidad del frente encontrado empeore significativamente.

En el análisis de los resultados no parece que se aplique ningún tipo de test estadístico que pueda dotar las conclusiones de confianza estadística. Ademas, se echa en falta alguna información sobre el algoritmo como el método implementado para gestionar las soluciones no dominadas en el archivo (o no está limitado en su tamaño? En este caso debería mencionarse también).

---

### Decision · Program_Chairs · 2025-03-20

**Decision:**

Accept

**Comment:**

De cara a preparar la versión camera ready, sugerimos a los autores que revisen detenidamente los comentarios de los revisores y que traten de mejorar el manuscrito según las recomendaciones de los revisores.